

# Interval Coded Scoring: a toolbox for interpretable scoring systems

Lieven Billiet[1,2], Sabine Van Huffel[1,2] and Vanya Van Belle[1]

[1] STADIUS Center for Dynamical Systems, Signal Processing and Data Analytics, Department of Electrical Engineering (ESAT), KU Leuven, Leuven, Belgium
[2] imec, Leuven, Belgium

## ABSTRACT

Over the last decades, clinical decision support systems have been gaining importance. They help clinicians to make effective use of the overload of available information to obtain correct diagnoses and appropriate treatments. However, their power often comes at the cost of a black box model which cannot be interpreted easily. This interpretability is of paramount importance in a medical setting with regard to trust and (legal) responsibility. In contrast, existing medical scoring systems are easy to understand and use, but they are often a simplified rule-of-thumb summary of previous medical experience rather than a well-founded system based on available data. Interval Coded Scoring (ICS) connects these two approaches, exploiting the power of sparse optimization to derive scoring systems from training data. The presented toolbox interface makes this theory easily applicable to both small and large datasets. It contains two possible problem formulations based on linear programming or elastic net. Both allow to construct a model for a binary classification problem and establish risk profiles that can be used for future diagnosis. All of this requires only a few lines of code. ICS differs from standard machine learning through its model consisting of interpretable main effects and interactions. Furthermore, insertion of expert knowledge is possible because the training can be semi-automatic. This allows end users to make a trade-off between complexity and performance based on cross-validation results and expert knowledge. Additionally, the toolbox offers an accessible way to assess classification performance via accuracy and the ROC curve, whereas the calibration of the risk profile can be evaluated via a calibration curve. Finally, the colour-coded model visualization has particular appeal if one wants to apply ICS manually on new observations, as well as for validation by experts in the specific application domains. The validity and applicability of the toolbox is demonstrated by comparing it to standard Machine Learning approaches such as Naive Bayes and Support Vector Machines for several real-life datasets. These case studies on medical problems show its applicability as a decision support system. ICS performs similarly in terms of classification and calibration. Its slightly lower performance is countered by its model simplicity which makes it the method of choice if interpretability is a key issue.

Corresponding authors
Lieven Billiet,
lieven.billiet@esat.kuleuven.be
Sabine Van Huffel,
sabine.vanhuffel@esat.kuleuven.be

## INTRODUCTION

The last decades have seen a wide growth of the application of Machine Learning and Data Science in healthcare, giving rise to the field of *health informatics*. The term itself was probably first used in the seventies (*Protti, 1995*). It can be described to be at 'the crossroad of information science, computer science, medicine, and healthcare, with a wide range of application areas including nursing, clinical care, public health, and biomedicine' (*Liang, 2010*). Largely, it has gained importance due to the increase of computational power, but also due to the enormous amount of medical data that has become available in an accessible digital format. This goes even as far as leading to *information overload*. It entails that the available information cannot be processed effectively by an observer (e.g., a clinician) (*Speier, Valacich & Vessey, 1999*). Hence, computer systems have been designed to serve as *Clinical Decision Support Systems* (CDSS) e.g., to improve early detection of specific problems. Several studies have shown that such systems indeed increase practitioner performance and improve patient outcome, while decreasing the costs at the same time (*Johnston et al., 1994*; *Garg et al., 2005*; *Chaudhry et al., 2006*). Meanwhile, different systems have been developed. One can roughly distinguish two types: data-based and knowledge-based decision support systems (*Berner & La Lande, 2016*).

### Data-based decision support

This type of CDSS is the most exemplary of the application of general machine learning techniques on (bio)medical problems. The underlying idea is harvesting the information contained in past data. Although some techniques can use the full database of previous records, the more usual approach is to extract a model that summarizes the most important discriminating information for the problem at hand using a wide variety of available techniques.

*Regression techniques* try to predict a continuous variable. One of the easiest formulations is least-squares linear regression which simply fits a linear model $\mathbf{w}^T X + b$ with weight vector $\mathbf{w}$, offset $b$ and data matrix $X$ by minimizing the quadratic error with regard to the target vector $\mathbf{y}$ i.e., $\min_w \|\mathbf{w}^T X + b - \mathbf{y}\|^2$. However, the optimization problem is often required to satisfy certain additional conditions, which can be obtained by introducing penalty terms. This is termed regularized least squares. One such condition is sparsity, a measure of model simplicity. The goal is to obtain a well-performing model by focusing on as few variables as possible. In other words, the aim is to have as many zeros in $\mathbf{w}$ as possible. It has become very popular with the advent of the LASSO (*Tibshirani, 1996*) and elastic net (*Zou & Hastie, 2005*). Both add an $\ell_1$-norm penalty on $\mathbf{w}$ to promote sparsity as a placeholder for the ideal $\ell_0$-norm which has undesirable optimization properties. The elastic net formulation imposes an additional ridge regression $\ell_2$-norm penalty to obtain a more stable result. The use of regression as a CDSS has largely been restricted to logistic regression for likelihood estimation (*Steyerberg et al., 2001*). Still, elastic net and LASSO have been used, if not as CDSS, to gain insight in several biomedical problems and as feature selection technique. At its introduction, the elastic net was immediately applied to microarray data due to its high dimensionality (*Zou & Hastie, 2005*). Similarly, (*Shen et al., 2011*) used it to identify neuroimaging and proteomic biomarkers for Alzheimer's

disease. More recently, *Ryali et al. (2012)* applied the elastic net to estimate functional connectivity from fMRI data, whereas *Greene et al. (2015)* exploited it to detect mobility parameters progression of early multiple sclerosis and to discriminate patients from a control population based on Timed-Up-and-Go (TUG) tests.

*Classification techniques* try to distinguish several data classes in either a binary or multiclass setting. Well-known approaches include Support Vector Machines (SVM) (*Vapnik, 1995*), Naive Bayes (NB) classifiers (*Lewis, 1998*), decision trees (DT) (*Quinlan, 1986*), random forests (RF) (*Ho, 1995*) and neural networks (NN) (*Bishop, 1995*). All of them have been applied to solve biomedical problems and serve as CDSSs. For example, all of them except NN were compared for the early detection of late-onset neonatal sepsis (*Mani et al., 2014*). RFs have been used for classification of microarray data (*Díaz-Uriarte & Alvarez de Andrés, 2006*). Another application involves screening for ECG arrhythmia using, among others, SVMs (*Martis, Chakraborty & Ray, 2014*). Further examples includes NNs for severity assessment of breast cancer (*Maity & Das, 2017*) and SVMs for detection of mild cognitive impairment and Alzheimer disease from multimodal brain imaging (*Shen et al., 2014*). A final example is the so-called 'doctor AI' system (*Choi et al., 2016*). It is a broad system using a recursive NN to predict patient diagnosis, among others.

The machine learning approaches outlined above have some clear advantages. They incorporate experience based on previous cases, refining it to its essence. Training is mostly automatic without the need for manual intervention. Moreover, the methods have solid mathematical and statistical foundations built on decades of extensive research and have been applied in many different fields. Finally, they also boast impressive results in almost all healthcare domains, sometimes surpassing clinical experts, given the improved outcome mentioned earlier.

The downside of many of these models is their lack of transparency and interpretability. Whereas one can still understand the outcome of a logistic regression model or a decision tree, this is much harder for nonlinear SVMs or NNs. Particularly in case of CDSSs aimed at providing a diagnosis or prognosis, such black box characteristics are far from appealing. On the one hand, the clinician has to trust the model. He or she treats patients according to its guidance, and does not always understand how the algorithm reached its conclusion. Hence, there is no easy way to check if the decision corresponds with personal or more general clinical experience. On the other hand, such black box decision-making CDSSs create a problem of (legal) responsibility. If a clinician does not fully understand the algorithm or its conclusion, can he or she be held responsible for carrying out the treatment?

## Knowledge-based decision support

Structured ways to summarize human knowledge rather than extracting it from data predate the data-based systems. The first computerized attempts were the *expert systems* based on early artificial intelligence. Researchers tried to represent human knowledge using, often incomplete, sets of rules. Decisions and explanations for new data could be obtained from inference engines. This lead to systems such as MYCIN for the selection of antibiotics or INTERNIST-I for diagnosis in internal medicine (*Duda & Shortliffe, 1983*). Many of

these systems could output their inference trace when requested. This allowed clinicians to see how the engine obtained its decision by reviewing which rules had been applied.

Another approach to summarize knowledge favours manual applicability. Such *medical scoring systems* have been used for decades and continue to be used and developed until today. Although conceptually similar expert systems, they are usually much simpler and have a more limited focus. They consist of several simple logic rules e.g., whether the patient's age is higher than a threshold. Each rule has a corresponding number of points that should be added if it holds true. Adding all these points yields a score, which can be linked to an empirical risk and possible treatment options. Examples of scoring systems include Alvarado for appendicitis (*Alvarado, 1986*), $CHA_2DS_2$-VASc for atrial fibrillation (*Lip et al., 2010*), Glasgow for pancreatitis (*Mounzer et al., 2012*), CURB-65 for pneumonia (*Jeong et al., 2013*) and ASDAS for disease activity in ankylosing spondylitis (*Lukas et al., 2009*).

The advantage of knowledge-based systems is the fact that they are often constructed based on accepted clinical knowledge and they are interpretable, particularly in case of scoring systems. However, they are often crude rules of thumb, albeit sometimes statistically verified post hoc. This is particularly true for expert and scoring systems that have been constructed through questionnaires and structured discussions.

## Data-driven interpretability

Some other approaches have tried to combine aspects of both data and knowledge-driven techniques. The majority focuses on the extraction of an interpretable view on either the data or an existing model. For example, several studies investigated the construction of rule systems from data. Sometimes rules are generated directly, e.g., using Bayesian techniques (*Letham et al., 2015*). Alternatively, one can construct them based on SVMs (*Martens et al., 2007*; *Barakat & Bradley, 2010*).

The Interval Coded Scoring (ICS) system discussed in this paper resides in this category as well (*Van Belle et al., 2012*). Based on mathematical optimization theory, it constructs a scoring system in a semi-automatic way. Hence, the training supervisor can steer the model towards the desired trade-off between model complexity and classification performance. In contrast to similar work by *Ustun & Rudin (2016)* it focuses on value intervals rather than variables, allowing for more fine-tuned models. Furthermore, it explicitly handles the concepts of main effects and variable interactions (*Billiet, Van Huffel & Van Belle, 2016*). Moreover, an efficient implementation allows it to process larger datasets (*Billiet, Van Huffel & Van Belle, 2017*). Finally, it stresses an appealing visualization of the model as a way to improve its interpretability.

This work introduces a toolbox for ICS. 'Methods' gives an overview of the functionality of the toolbox, including a summary of the theoretical background. However, the main focus is on the accessibility provided by the toolbox interface and the range of applications it facilitates. Therefore, 'Results and Discussion' shows outcomes for various publicly available real-life medical datasets and makes a comparison with standard machine learning algorithms discussed earlier. Finally, the last Section provides some conclusions. The software is made available for academic research at https://www.esat.kuleuven.be/stadius/icstoolbox.

## METHODS

This section covers three topics. Firstly, the Interval Coded Scoring algorithms are described. Secondly, the toolbox itself is discussed. Lastly, details of the toolbox validation approach are provided.

**A note on mathematical notation.** A scalar constant $N$ will be typeset as uppercase, whereas a variable $a$ is indicated by lowercase notation. Bold typesetting indicates a vector $\mathbf{a}$ (lowercase) or a matrix $\mathbf{A}$ (uppercase). Additionally, subscripts as in $a_i$ indicate the $i$th scalar element of a vector $\mathbf{a}$. Finally, superscripts will sometimes be used as a grouping index. For example, as explained later, $\tau_i^p$ represents the $i$th threshold on the $p$-th variable. It should be clear from the context if exponentiation is meant instead. Moreover, apart from the usual notation for the real numbers $\mathbb{R}$, the set of binary numbers $\{0, 1\}$ will similarly be referred to as $\mathbb{B}$.

### Interval coded scoring

In essence, ICS is a general binary classifier. It starts from data $\mathbf{X} \in \mathbb{R}^{N \times N_d}$ with $N$ observations of dimensionality $N_d$, paired with a vector $\mathbf{y}$ containing class labels $y_i \in \{-1, 1\}$ ($i = 1, \ldots, N$). In the end, it produces a model $\varphi(\mathbf{X})\mathbf{s}$ with $\mathbf{s}$ a sparse integer vector of points and $\varphi(.)$ a data transformation. This model can be obtained by two different formulations of the core algorithm: (i) lpICS, based on linear programming, and (ii) enICS, based on the elastic net. Nevertheless, the general flow is the same for both approaches, as outlined in pseudocode in Algorithm 1.

---

**Algorithm 1** General overview of ICS. Detailed explanation is provided below.

---

1: Construct $\tau^p \quad \forall x_p, \quad p \in [1, N_d]$
2: $\mathbf{Z} \leftarrow \varphi(\mathbf{X}, \tau)$
3: $\mathbf{w} \leftarrow \mathrm{ICS}(\mathbf{Z})$
4: select $a$
5: $\mathbf{w} \leftarrow \mathrm{ICS}(\mathbf{Z}, a)$ {see Algorithm 2 and 3}
6: $\mathbf{w}, \mathbf{Z} \leftarrow \mathrm{nonEmptyVars}(\mathbf{w}, Z)$
7: **repeat**
8: $\quad n_{old} \leftarrow \#\mathrm{vars}(\mathbf{w})$
9: $\quad$ select $a$
10: $\quad \mathbf{w} \leftarrow \mathrm{ICS}(\mathbf{Z}, a)$ {see Algorithm 2 and 3}
11: $\quad \mathbf{w}, \mathbf{Z} \leftarrow \mathrm{nonEmptyVars}(\mathbf{w}, \mathbf{Z})$
12: **until** $n_{old} == \#\mathrm{vars}(\mathbf{w})$
13: $\mathbf{s} \leftarrow \mathrm{scaleAndRound}(\mathbf{w})$
14: risk profile $\leftarrow \mathrm{logRegress}(\mathbf{Z}\mathbf{s}, \mathbf{y})$

---

Line 1 and 2 represent the data transformation. Here, we only discuss the shape of the data after transformation, not its content (data encoding). This differs depending on the core algorithm, as will be discussed in sections 'Linear Programming ICS (lpICS)' and 'Elastic net ICS (enICS)'.

The variables in data $\mathbf{X}$ can be binary, ordinal, categorical or continuous. A new binary dataset $\mathbf{Z}$ is constructed by dividing the range of the original variables in a number of intervals through thresholds $\tau$. These intervals are defined as follows:

- **Binary observation** $x$ is expanded to two binary values, one corresponding to 0 and one corresponding to 1. Hence, $x \in \mathbb{B} \rightarrow \mathbb{B}^2$.
- **Categorical observation** $x$ is expanded into a number of binary values equal to the number of categories $N_c$: $x \in \{p_1, p_2, \ldots, p_{N_c}\} \rightarrow \mathbb{B}^{N_c}$ with $p_k$ ( $k \in 1, \ldots, N_c$) a representation of category $k$.
- **Continuous observation** $x$ results in $N_r$ binary values through thresholds $\tau_i^p$ with $i = 1, \ldots, N_r - 1$. $N_r$ can be set beforehand for each variable $x^p$. The thresholds are calculated as to divide the range of $x^p$ into $N_r$ equal percentiles (intervals) with the restriction that any percentile must contain at least five data points. Otherwise, the number of intervals is reduced. For each of the intervals, a new binary variable is created. Therefore, $x \in \mathbb{R} \rightarrow \mathbb{B}^{N_r}$.
- **Ordinal observations** are treated as continuous data, except that thresholds can only be integer values.
- **Interactions** can be modelled by combining the principles above. For example, the interaction between heart rate and energy readings from accelerometry can indicate whether an increased heart rate is due to physical activity or not. Imagine an observation of these two continuous variables $[x^1, x^2]$, with a number of intervals $N_r^1$ and $N_r^2$, respectively. This single observation would be expanded to a binary matrix: $[x^1, x^2] \in \mathbb{R}^2 \rightarrow \mathbf{B} \in \mathbb{B}^{N_r^1 \times N_r^2}$.

To summarize, every variable $x^p$ is converted to a binary vector, an interaction expands to a binary matrix. A full observation $\mathbf{x}_i$ covering all variables is therefore converted to the concatenated vector $\mathbf{z}_i$. In case of interactions, matrices are vectorized before concatenation. All transformed observations $\mathbf{z}_i$ are gathered in the matrix $\mathbf{Z}$.

Line 3 calls the core algorithm (`lpICS` or `enICS`). It returns a real-valued sparse vector $\mathbf{w}$ containing weights corresponding to all binary variables in $\mathbf{Z}$.

The model so far can be tuned further in lines 4–5 through *iterative reweighting*. Its goal is to yield an even simpler model, as shown in Fig. 1. Consider a dataset with a single continuous variable, that is, $\mathbf{X} \in \mathbb{R}^N$ is a vector of $N$ observations. It has been expanded to a binary representation with e.g., 18 intervals $\mathbf{Z} \in \mathbb{B}^{N \times 18}$ as explained earlier. Through execution of line 3, weights $\mathbf{w}$ have been obtained, shown in full gray line in the Figure. However, it shows many small weight changes. In contrast, a model capturing only the main changes is more desirable. Firstly, it is more likely to capture general tendencies rather than sample-specific changes. Secondly, it is simpler to understand. The desired simplification can be achieved by reweighting as shown with the dashed line in the Figure. The user can interact by selecting the reweighting constant $a$, which influences the trade-off between model complexity and performance. The specific reweighting formulation will be discussed for each method separately in the next subsections.

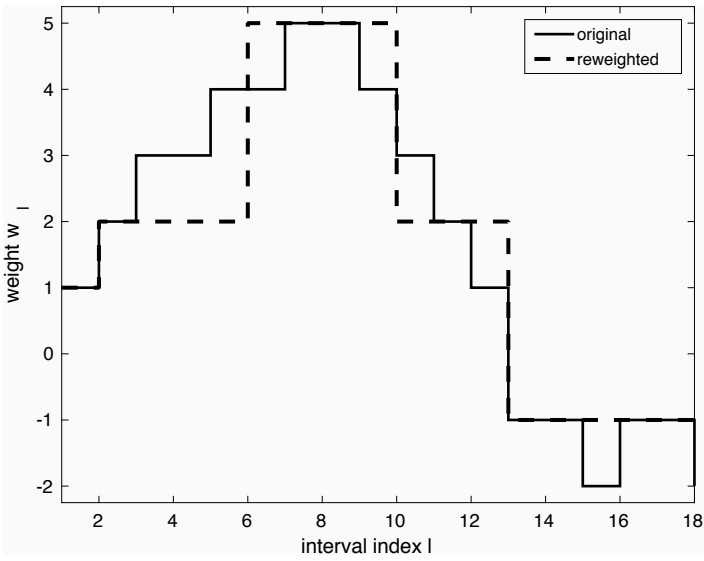

**Figure 1** **Simplification of a model via iterative reweighting.**

If all weights for the intervals of an original variable $x^p$ are zero, this variable (all of its binary intervals) can be eliminated. Hence, ICS has built-in feature selection. The simplification of the problem through removal of variables with zero coefficients is carried out in line 6.

Lines 7–12 show a loop containing lines 3–6. Here, the model training is repeated with the selected variables until a stable model is obtained.

The resulting weight vector **w** is not necessarily integer-valued. Hence, line 13 converts it to integer score points **s** by scaling and rounding. Furthermore, this step simplifies the model by joining adjacent intervals with equal score points in case of continuous and ordinal variables.

The last line applies logistic regression to convert the scores to a risk profile, that is, it attributes a risk value to every score value.

### Linear Programming ICS (lpICS)

This ICS core approach is based on the soft-margin SVM formulation (*Vapnik, 1995*). It can be restated and solved efficiently as a linear programming problem. Normally, MATLAB's solver is used, but IBM's CPLEX is selected if available, since it drastically improves execution time. The `lpICS` formulation is presented in matrix notation in Eq. (1). The same symbols as in Algorithm 1 have been used, with a tilde to refer to its specific use in `lpICS`.

$$\begin{aligned} \min_{\tilde{\mathbf{w}}, b, \boldsymbol{\varepsilon}} \quad & \|\mathbf{D}\tilde{\mathbf{w}}\|_1 + \gamma \boldsymbol{\varepsilon}^T \mathbf{1}, & & \mathbf{D} \in \mathbb{R}^{N_{ds} \times N_s}, \tilde{\mathbf{w}} \in \mathbb{R}^{N_s} \\ s.t. \quad & \mathbf{Y}(\tilde{\mathbf{Z}}\tilde{\mathbf{w}} + b) \geq \mathbf{1} - \boldsymbol{\varepsilon}, & & \mathbf{Y} \in \mathbb{R}^{N \times N}, {}^{\mathrm{lp}}\tilde{\mathbf{Z}} \in \mathbb{B}^{N \times N_s}. \\ & \boldsymbol{\varepsilon} \geq \mathbf{0}, & & \boldsymbol{\varepsilon} \in \mathbb{R}^N \end{aligned} \tag{1}$$

Similarly to the original SVM formulation, it tries to fit a model $\tilde{\mathbf{Z}}\tilde{\mathbf{w}} + b$ on the class labels **y** in the diagonal matrix **Y**. The data matrix $\tilde{\mathbf{Z}}$ contains the binary transformed

data, yielding $N$ observations of $N_s$ intervals. Furthermore, the constraint enforces correct classification, allowing slack $\varepsilon$ with respect to the classification boundary. In the usual SVM objective function, the total slack is balanced against the classification margin through a hyperparameter $\gamma$, but `lpICS` minimizes $\|\mathbf{D}\bar{\mathbf{w}}\|_1$ instead. This penalty term corresponds to *Total Variation Minimization* (*Rudin, Osher & Fatemi, 1992*). The binary matrix $\mathbf{D}$ encodes $N_{ds}$ differences between coefficients of adjacent intervals of the same variable $x^p$ or variable interaction $(x^{p_1}, x^{p_2})$. In the latter case, both the horizontal and vertical differences in the unvectorized matrix shape are taken into account. Hence, the sparsity-inducing $\ell_1$ norm is applied on the difference of the weights. As a result, `lpICS` favours piecewise constant models as depicted in Fig. 1 because they have a sparse difference vector, or, equivalently, identical coefficients for adjacent intervals.

**Data transformation.** As mentioned in Algorithm 1, the matrix $\tilde{\mathbf{Z}}$ is the result of a transformation of the original data $\mathbf{X}$ guided by thresholds $\tau$. The `lpICS` transformation $\tilde{\varphi}$ supports both main effects (only a single variable $x^p$ is affected) and interactions $(x^{p_1}, x^{p_2})$. The transformation of a single observation $\mathbf{x}_i$ is given as:

$$
\begin{aligned}
&\tilde{\mathbf{z}}_i = [\tilde{\mathbf{z}}_i^1, \tilde{\mathbf{z}}_i^2, \ldots, \tilde{\mathbf{z}}_i^{N_d}, \tilde{\mathbf{z}}_i^{1,2}, \ldots, \tilde{\mathbf{z}}_i^{N_d-1, N_d}] \\
&\text{where} \quad \tilde{\mathbf{z}}_i^p = \tilde{\varphi}^p(x_i^p) \quad \text{with} \\
&\qquad \tilde{\varphi}^p(x_i^p) = [I(x_i^p < \tau_1^p), I(\tau_1^p \le x_i^p < \tau_2^p), \ldots, I(\tau_{N_p-1}^p \le x_i^p)] \\
&\text{and} \quad \tilde{\mathbf{z}}_i^{p_1, p_2} = \text{vec}(\tilde{\varphi}^{p_{1,2}}(x_i^{p_1}, x_i^{p_2})) \quad \text{with} \\
&\qquad \tilde{\varphi}_{k,l}^{p_1, p_2}(x_i^{p_1}, x_i^{p_2}) = I(\tau_{k-1}^{p_1} \le x_i^{p_1} < \tau_k^{p_1} \quad \& \quad \tau_{l-1}^{p_2} \le x_i^{p_2} < \tau_l^{p_2}) \\
&\qquad k \in \{1, \ldots, N_{p_1}\}, l \in \{1, \ldots, N_{p_2}\} \\
&\qquad \text{defining} \quad \tau_0^{p_1} = \tau_0^{p_2} = -\infty \quad \text{and} \quad \tau_{N_{p_1}}^{p_1} = \tau_{N_{p_2}}^{p_1} = \infty
\end{aligned}
\tag{2}
$$

In this equation, $I$ is an indicator function, yielding 0 or 1 when its argument is `false` or `true`, respectively. $N_p$ refers to the number of intervals for variable $x^p$. Summarizing, every observation $x_i^p$ of a variable $x^p$ is expanded into a binary vector $\tilde{\mathbf{Z}}_i^p$ containing only a single 1 indicating the interval to which the value of $x^p$ belongs. Two-way interactions $(x^{p_1}, x^{p_2})$ yield a binary matrix containing only a single 1. The `vec` in the fourth line in the equation indicates vectorization of this matrix to obtain a vector. Finally, transformations for several variable values are simply concatenated to obtain one sparse vector $\tilde{\mathbf{Z}}_i$, a row of the matrix $\tilde{\mathbf{Z}}$. Which main effects and interactions are considered initially can be set by the user.

**Reweighting.** To obtain the simplification shown in Fig. 1, iterative reweighting is performed. It is a repetitive execution of the `lpICS` formulation in Eq. (1), where $\mathbf{D}\bar{\mathbf{w}}$ has been replaced by $\tilde{\boldsymbol{\chi}}\mathbf{D}\bar{\mathbf{w}} \cdot \tilde{\boldsymbol{\chi}} \in \mathbb{R}^{N_{ds} \times N_{ds}}$ is a diagonal matrix containing weights $\tilde{\chi}_{i,i}$, $i = 1, \ldots, N_{ds}$. The reweighting values are computed based on the current solution of the problem $\bar{\mathbf{w}}$ as $\tilde{\chi}_{i,i} = 1/(\epsilon_1 + a|(\mathbf{D}\bar{\mathbf{w}})_i|)$. The reweighting can be summarized as:

---

**Algorithm 2** Iterative reweighting in `lpICS`.

1: Given: $\tilde{\mathbf{w}}$, $a$
2: **repeat**
3: $\quad \tilde{\mathbf{w}}^{\text{old}} = \tilde{\mathbf{w}}$
4: $\quad \tilde{\chi}_{i,i} = \frac{1}{\epsilon_1 + a|(\mathbf{D}\tilde{\mathbf{w}}^{\text{old}})_i|}, \forall i$
5: $\quad \tilde{\mathbf{w}} \leftarrow$ solve Equation 1 with $\mathbf{D}\tilde{\mathbf{w}}$ replaced by $\tilde{\chi}\mathbf{D}\tilde{\mathbf{w}}$
6: **until** AVG$(|\tilde{\mathbf{w}} - \tilde{\mathbf{w}}^{\text{old}}|) < \epsilon_2$ or #iter == 10

---

$\epsilon_1$ and $\epsilon_2$ are small values of 5e-4 and 1e-8, respectively. They help to avoid singularities and numerical issues. The value $a$ can be selected by the user based on crossvalidation results on the training data. The loop continues until convergence, with a maximum of 10 iterations. Note also the difference with the loop in Algorithm 1 which surrounds the call to Algorthm 2. In the formercase, the aim was to remove variables, whereas here, the goal lies in the simplification of the model for every variable. Yet, in extreme cases, this simplification of the variable might also result in its removal.

### Elastic net ICS (enICS)

The dimensionality of $\mathbf{Z}$ can be very high due to the binary expansion. As a consequence, the execution time is high, even though some solvers are able to take advantage of the sparse structure. To improve the execution time for larger problems, ICS supports a slightly different formulation based on elastic net, `enICS`. As in `lpICS` the sparsity is not induced on $\mathbf{w}$, but on the difference $\mathbf{Dw}$. However, this is done implicitly. Consider the formulation, in terms of the notation of `lpICS`:

$$\min_{\tilde{\mathbf{w}},b} \|\tilde{\mathbf{Z}}\tilde{\mathbf{w}} + b - \mathbf{y}\|_2^2 + \epsilon\|\mathbf{D}\tilde{\mathbf{w}}\|_2^2 \quad \text{s.t.} \quad \|\mathbf{D}\tilde{\mathbf{w}}\|_1 \leq t. \tag{3}$$

In this equation, the symbols have the same meaning and dimensions as before. Additionally, $\epsilon$ is a parameter fixed at 0.05. It provides the stability of elastic net to the LASSO formulation, but other than that, it can be ignored. The hyperparameter $t$ defines the tradeoff between sparsity (model simplicity) and performance. Note that the modelling is posed as regression rather than classification. Instead of minimizing the errors with respect to the classification border, it explicitly aims at the minimization of the deviation from the label values 1 and $-1$.

The advantage of using this approach becomes apparent when using a slightly different data representation converting it to a standard elastic net formulation, the actual `enICS`:

$$\min_{\hat{\mathbf{w}},b} \|\hat{\mathbf{Z}}\hat{\mathbf{w}} + b - \mathbf{y}\|_2^2 + \epsilon\|\hat{\mathbf{w}}\|_2^2 \quad \text{s.t.} \quad \|\hat{\mathbf{w}}\|_1 \leq t. \tag{4}$$

Conceptually, $\hat{\mathbf{Z}} = \tilde{\mathbf{Z}}\mathbf{D}^{-1}$, although it is never calculated explicitly, but results from direct construction. Therefore, Eq. (4) directly optimizes the weight differences $\hat{\mathbf{w}}$. *Zhou et al. (2015)* have shown that the elastic net is equivalent to the dual formulation of a squared-hinge loss support vector machine. Due to its formulation, it can be efficiently solved in the primal (*Chapelle, 2006*) with a complexity related to the number of data points rather than the size of the expanded feature space, which is usually greater. It makes the problem complexity independent of the size of the feature space and allows for much faster

execution. Furthermore, it automatically solves either the primal problem or the dual problem, depending on its characteristics e.g., in case of a large dataset. This approach is called SVEN (Support Vector Elastic Net).

**Data transformation.** It was mentioned that $\hat{\mathbf{Z}}$ is the result of direct construction from the data, similar to $\tilde{\mathbf{Z}}$. Hence, an `enICS` data transformation $\hat{\varphi}$ can be defined as follows for a single observation $\mathbf{x}_i$:

$$
\begin{aligned}
\hat{\mathbf{Z}}_i &= [\hat{\mathbf{Z}}_i^1, \hat{\mathbf{Z}}_i^2, \ldots, \hat{\mathbf{Z}}_i^{N_d}] \\
\text{where} \quad \hat{\mathbf{Z}}_i^p &= \hat{\varphi}^p(x_i^p) = [1, I(x_i^p \geq \tau_1^p), I(x_i^p \geq \tau_2^p), \ldots, I(x_i^p \geq \tau_{N_p-1}^p)]
\end{aligned}
\tag{5}
$$

Notice how the transformation uses the same bins as `lpICS` with the same bin thresholds $\boldsymbol{\tau}$. However, $\hat{\varphi}$ uses more than a single 1 to encode a data point. If a value corresponds to a certain interval, all previous intervals are filled with 1. Hence, the weights no longer reflect the impact of an interval, but a difference with respect to the previous interval. Of course, in the end, one is still interested in the importance of the intervals themselves. This is easily calculated from the `enICS` weights by cumulative summing for each variable encoded in a reconstruction matrix $\mathbf{R}$, such that $\tilde{\mathbf{w}} = \mathbf{R}\hat{\mathbf{w}}$.

Notice also how Eq. (5) no longer contains interaction terms. Indeed, SVEN is not applicable in that case since Eq. (3) becomes an arbitrary *generalized elastic net*. This can be converted to a standard elastic net, but additional constraints need to be added (*Xu, Eis & Ramadge, 2013*) which makes SVEN impossible. Therefore, `enICS` only supports main effects.

**Reweighting.** The reweighting procedure for `enICS` is similar to `lpICS`. Based on the current solution $\hat{\mathbf{w}}$ and the weighting parameter $a$ supplied by the user, reweighting can be done as expressed in Algorithm 3. A key difference with Algorithm 2 lies in the calculation and application of the reweighting values $\hat{\chi}$ since they are applied on the data $\hat{\mathbf{Z}}$, rather than in the (`lpICS`) regularization on $\mathbf{D}\tilde{\mathbf{w}}$. Hence, they are the reciprocal of the `lpICS` reweighting values $\tilde{\chi}$, neglecting $\epsilon_1$. The value of $\epsilon_2$ remains at 1e-8.

---

**Algorithm 3** Iterative reweighting in `enICS`.

---

1: Given: $\hat{\mathbf{w}}$, $a$
2: **repeat**
3:     $\hat{\mathbf{w}}^{\text{old}} = \hat{\mathbf{w}}$
4:     $\hat{\chi}_{i,i} = a|\hat{w}_i|$
5:     $\hat{\mathbf{w}} \leftarrow$ solve Equation 4 with $\hat{\mathbf{Z}}$ replaced by $\hat{\mathbf{Z}}\hat{\chi}$
6: **until** $\text{AVG}(|\hat{\mathbf{w}} - \hat{\mathbf{w}}^{\text{old}}|) < \epsilon_2$ or #iter == 25

---

### Preselection

`enICS` was introduced as a way to improve the execution speed of ICS for larger datasets with a higher dimensionality. Instead of using another model, one can also perform feature selection beforehand (*Billiet, Van Huffel & Van Belle, 2017*). This reduces the dimensionality of the data. Yet, the aim is still to take ICS's binning structure into account. Therefore, the so-called *preselection* is performed as a four-step procedure:

1. The data transformation in Eq. (2) is performed on the dataset $\mathbf{X}$. Note that interaction can be included at this point as well. Using the computed $\tilde{\mathbf{Z}}$ and the known labels, one can solve a standard Support Vector Machine classification problem with linear kernel yielding coefficients $\boldsymbol{\omega}$.

2. A new data set $\mathbf{X}_2$ is constructed. Main effects are modelled as $\mathbf{X}_2^p = \tilde{\mathbf{Z}}^p \boldsymbol{\omega}^p$, $\forall\, p = 1, \ldots, N_d$. This represents the variable values in $\mathbf{X}$ by the coefficients of their intervals. Hence if only main effects are involved, $\mathbf{X}_2 \in \mathbb{R}^{N \times N_d}$ has the same size as $\mathbf{X}$. Each considered interaction increases the dimensionality of $\mathbf{X}_2$ with one. These additional columns are given as $\tilde{\mathbf{Z}}^{p_1,p_2} \omega^{p_1,p_2}$, one for each interaction between variables $x^{p_1}$ and $x^{p_2}$. This approach can be considered as a data-driven discrete non-linear transformation. $\mathbf{X}_2$ encodes data trends for each variable and interaction.

3. The goal remains to eliminate certain variables. To obtain this, any feature selection method could be applied on the trend-informed data $\mathbf{X}_2$. Here, the elastic net was chosen. Similarly to the iterative reweighting discussed earlier, the selection is semi-automatic: the user can decide on the sparsity based on cross-validated performance estimation.

4. Finally, the elastic net indicates which columns of $\mathbf{X}_2$ should be kept. Since every column corresponds to one main effect or interaction, these columns correspond to the set of effects and interactions that ICS should consider. This is an implicit input in the transformation function $\varphi$ in the main ICS flow in Algorithm 1.

## The ICS toolbox

The previous subsection discussed the data transformations, the two different models `lpICS` and `enICS`, the use of preselection and also the general ICS algorithm. All functionality is implemented as a MATLAB toolbox. This toolbox also provides an interface layer for easy access. It includes a way to describe a problem and specify options, a graphical interaction method for preselection and iterative reweighting and a clear visualization of the model with highlighting of several assessment parameters. The toolbox also contains a manual with examples for further reference.

The current subsection describes the ICS toolbox. It shows how to set up a problem, how user interaction is implemented and how a model can be visualized and assessed.

### *Setting up a problem*

The toolbox groups all information about a specific ICS problem in a MATLAB struct *problem*. It has only two mandatory fields: *xtrain*, containing the training data and *ytrain*, the corresponding (binary) labels. The latter can contain any two numerical values, which will be transformed to the set $\{-1, 1\}$. Similarly, one can supply test data in the fields *xtest* and *ytest*. Checks on dimensionality correspondence are performed automatically, yielding error messages for incorrect input. Other fields that can be set include the variable types, their names, the groups (which main effects and/or interactions should be considered) and the algorithm to be used (`lpICS` or `enICS`). Furthermore, an options field allows to access more advanced settings such as whether the final model should be visualized, whether preselection should be performed, the maximum number of intervals for each variable,

whether selection of the reweighting parameter should be automatic and if so, which threshold should be used to do so. The details of all the fields are discussed extensively in the ICS manual accessible with the toolbox at https://www.esat.kuleuven.be/stadius/icstoolbox.

Once a problem has been defined, the algorithm can be started by calling the *buildICS* function with the problem struct as argument. Snippet 1 shows a minimal and full example to get started with ICS. It assumes the existence of a file *data.mat* containing training matrices $\mathbf{X}_{tr}$, $\mathbf{y}_{tr}$ and test matrices $\mathbf{X}_{te}$, $\mathbf{y}_{te}$. The matrices $\mathbf{X}_{tr}$ and $\mathbf{X}_{te}$ contain five variables (columns). Note that part of the code in the full setup is superfluous, since it corresponds to the defaults. It is only included as illustration.

Snippet 1: A minimal and full ICS setup example

```
load data.mat; %contains Xtr, Xte, ytr and yte

%%%%% minimal setup %%%%%
%set training data fields
problem1.xtrain = Xtr;
problem1.ytrain = ytr;

%%%%% full setup %%%%%
%set data fields
problem2.xtrain = Xtr;
problem2.ytrain = ytr;
problem2.xtest = Xte;
problem2.ytest = yte;

%the numbers refer to their respective columns in Xtr and Xte
%x1 and x3: continuous, x2 and x4: categorical, x5: binary
problem2.var = struct('cont',[1,3],'cat',[2,4],'bin',[5],'ord',[]);
problem2.varnames = {'foo','bar','rock','paper','scissors'};
%main effects for variables x1, x2 and x5,
%interactions between variables x1 and x3, x2 and x4
problem2.groups = {1, 2, 5, [1,3], [2,4]};
%using lpics
problem2.mainmethod = 'lp';

%options
problem2.options.show = true;
problem2.options.preselect = true;
%the maximum number of intervals for all variables is set to 42
%in practice, this only refers to continuous and ordinal variables
%categorical and binary variables revert to the defaults.
problem2.options.maxbins = 42;
%manual tuning for preselection and reweighting
problem2.options.auto = false;
%suggested weights will result in an AUC of 75% of the maximum cross
%validation AUC for preselection and 80% for reweighting
problem2.options.cutoff = [0.75, 0.8];

%%%%% compute the ICS model for both problems %%%%%
ICS1 = buildICS(problem1);
ICS2 = buildICS(problem2)
```

### Interactive weight selection

Users can interact with the training procedure at two points. Firstly, to make a sparsity-performance trade-off for preselection. Secondly, to balance the model complexity with regard to performance and risk calibration during the iterative reweighting. Only the latter is discussed here as they are very similar.

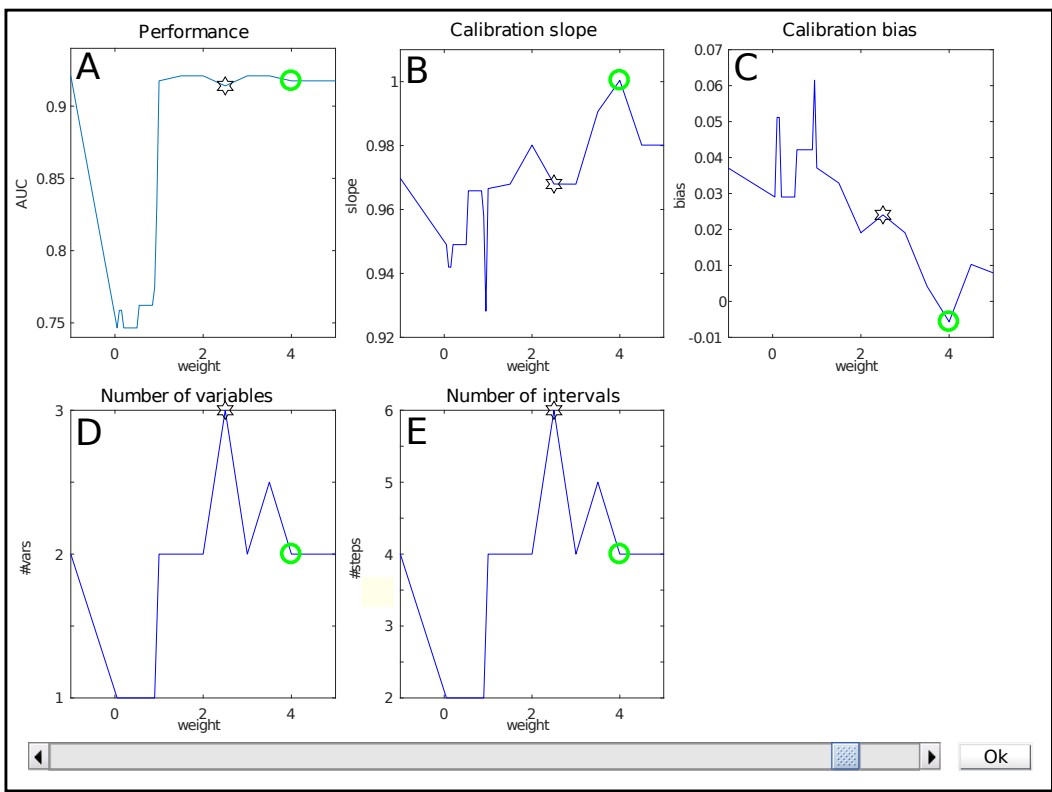

**Figure 2** **The weight selection interface during iterative reweighting shows crossvalidation results for various values of the reweighting parameter ('weight') *a*.** The model performance is expressed by the Area under the ROC curve (A), calibration is assessed using the slope and bias of the calibration curve (resp B, C). The complexity of the model is indicated by the number of variables (D) and intervals (E). Selection of a weight *a* occurs via the slider at the bottom.

The interface for weight selection is shown in Fig. 2. It presents five graphs with the weight as independent variable. The results shown are based on cross-validation on the training data.

The top row shows performance measures. Figure 2A contains the Area under the ROC curve (AUC) as general performance characteristic. It equals 1 in case of perfect classification and 0.5 for guessing. The other two graphs in the top row represent a summary of the calibration curve relating the estimated risk with the empirical risk, namely as slope and bias (Fig. 2B , resp. Fig. 2C). In an ideal case, it has a unit slope and zero bias.

The bottom row indicates model complexity. Figure 2D lists the number of variables for each weight, whereas Fig. 2E displays the total number of intervals. Often, these two are closely related, but this might differ e.g., when interactions are present in the model.

Each graph shows a star representing the estimated optimal weight based on the user-defined cutoff. However, in this case it is clear that the model can be improved manually. The selection of the user is indicated with a green circle. Here, it indicates a model with a lower estimated complexity, higher AUC and better calibration values.

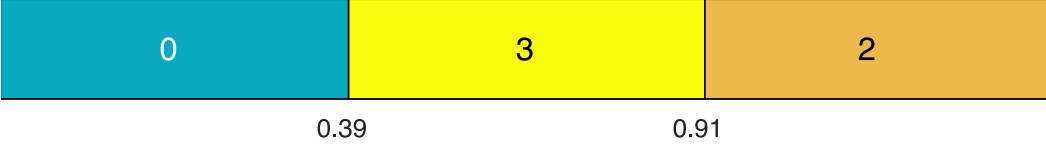

**Figure 3  An example of a main effect with points for each interval indicated by colours and numbers.** The thresholds $\tau$ are visible at the interval boundaries.

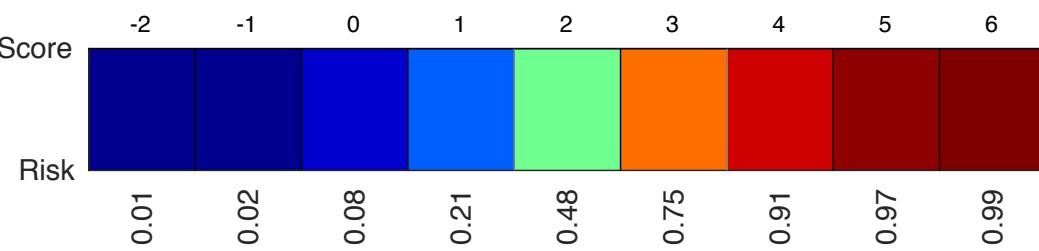

**Figure 4  An example of a risk profile with the scores on top and the corresponding risks at the bottom.**

The user can select the preferred weight from the evaluated discrete set by means of a slider. A special value is $-1$. It corresponds to no reweighting.

### Visualization

Visualization is an important factor. The toolbox supplies both a colour and grayscale representation. The visualization of the model itself consists firstly of a representation of all effects. An example for a main effect with three intervals is displayed in Fig. 3. The values of the thresholds $\tau$ are marked at the interval boundaries. The number of points for each interval is indicated by a colour code and by numerical values. In this case, if the value under consideration is higher than 0.91, two points will be added to the total score. Similarly, this can be done for other effects not displayed here. The same colour scheme is used for all effects, that is, the same colours represent the same values. In case of interactions, the representation is a grid rather than a row.

A second element of the model is the colour-coded risk profile, as shown in Fig. 4. It maps the scores (sum of the points) on top to estimated risks at the bottom. For example, if the sum of the contributions of all effects in the model yield a final score of three, the risk of belonging to the target class would be 75%.

The toolbox provides a listing of all effects and options for its visualization by means of a graphical interface (Fig. 5). It is automatically called at the end of model training (unless disabled), but can also be called on an already trained model with the function *showICS*. The Figure shows the model for the *ionosphere* dataset included with MATLAB. Complete code to generate it is available as a demo file in the toolbox. The two buttons on top display the assessment module (discussed below) and the risk profile. The middle part displays all selected effects. Here, three effects are part of the final model, as can be seen in the listbox. To its right, one can indicate if numerical values should be overlaid and whether to use

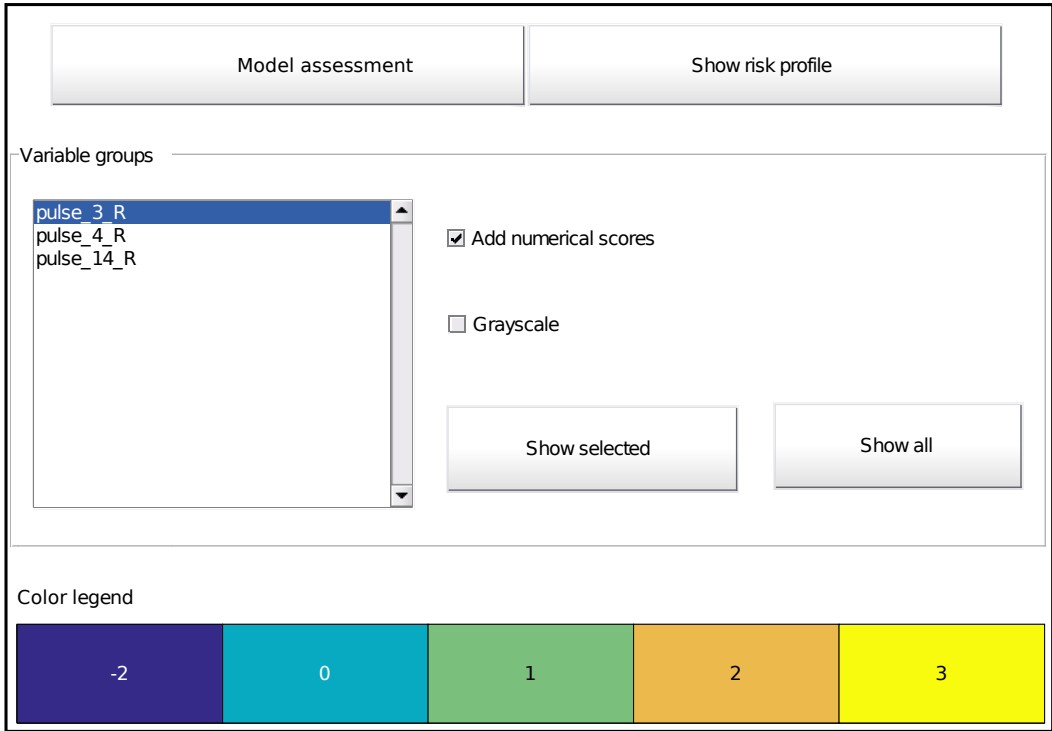

**Figure 5** **Main graphical interface to access the model. Two buttons at the top display the assessment interface (Fig. 6) or the risk profile (Fig. 4), respectively.** The middle part shows all effects in the model as a list and buttons to display the selected effect or all effects at once (Fig. 3). Optionally, one can overlay the interval points or use a grayscale representation. The bottom part provides the colour legend common to all displayed effects.

colour coding or grayscale (e.g., for printing). The model details for one or more effects can be displayed via the buttons. Finally, the bottom part contains the common colour legend for all effects.

### Assessment

The assessment module can be launched from the visualization interface (see Fig. 5). Normally, it shows assessment results for the training data only. If test data has been included at model setup, test results are summarized as well. Additionally, a model can be applied on new data with the function *applyICS*. Its outputs can be given as additional inputs to *showICS* to include results on the new data in the assessment.

The assessment window is displayed in Fig. 6. It evaluates the model at two levels. Firstly, the two graphs at the top show an alternative representation of the risk profile (Fig. 6A) and the calibration curve (Fig. 6B). The latter evaluates how well the predicted risk matches the empirical risk. Numerically, this is summarized in the lower right quadrant with the slope and bias of the curve, for training and test data. Secondly, the classification performance is evaluated using the Receiver Operating Characteristic (ROC) curve (Fig. 6C). It shows the tradeoff between sensitivity and specificity with an indication of the chosen tradeoff point. Moreover, the Area Under the Curve (AUC) is a robust indication of performance. Hence,

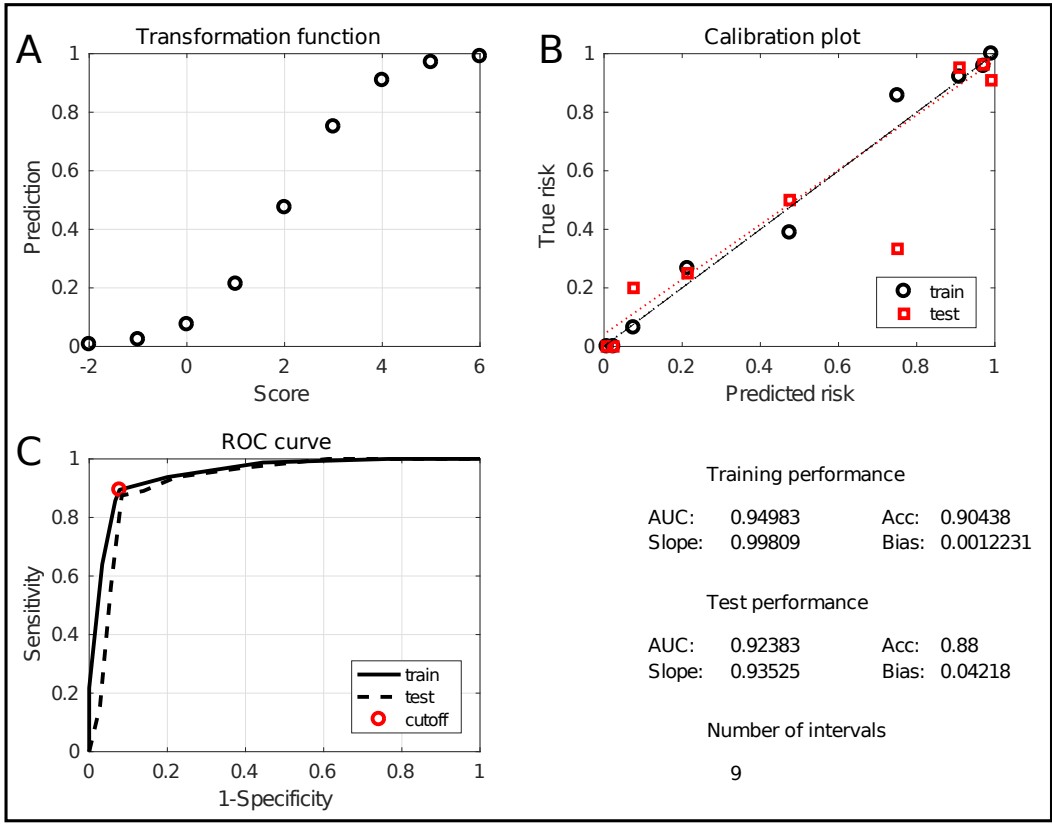

**Figure 6** **The ICS assessment interface.** It shows the transformation between scores and predictions (A), the calibration curve (B) and the ROC curve (C). It also evaluates the performance and complexity numerically by means of the AUC, the accuracy, the slope and bias and the total number of intervals.

it is included in the summary in the lower right quadrant, together with the classification accuracy (percentage of correctly classified samples). Finally, the number of intervals is included as a measure for the model complexity. The combination of these measures allows to thoroughly assess the quality of the obtained model.

## Toolbox validation

The effectiveness of the toolbox is validated by its application to a range of real-life datasets taken from the Machine Learning repository of the University of California, Irvine (UCI) (*Lichman, 2013*). They will be discussed briefly in the following paragraphs. All of the considered datasets are medical. Nevertheless, the usefulness of ICS is not restricted to this domain. Many other areas could also benefit from interpretable models e.g., process management, behaviour studies or modelling of physics systems. ICS will be run with its standard settings (automatic tuning) in four setups: `lpICS` with and without preselection and `enICS` with and without preselection. The validation also includes a performance comparison with other classification techniques. Naive Bayes and linear and nonlinear (gaussian kernel) Least-Squares Support Vector Machines (*Suykens, Van Gestel & De Brabanter, 2002*) have been chosen as they are widely accepted techniques. Decision

Trees (DT) have been included as an alternative interpretable classifier. Stratified random sampling selects two-thirds of the data for training and one-third for testing. In case of binary data, Naive Bayes builds the model using multivariate multinomial distributions instead of the default normal distributions. The SVMs are tuned via simulated annealing, whilst the Decision Trees are pruned to obtain the tree with the smallest cross-validation classification loss on the training data.

**Acute inflammations** (*Czerniak & Zarzycki, 2003*)**.** The set contains 120 observations with 6 variables, one continuous and five binary. It was originally intended to test a new expert system for diagnosis of two diseases of the urinary system, namely acute inflammations of urinary bladder and acute nephritis. As such, the set defines two classification problems: each disease versus a control group. The set contains 30 healthy controls, 40 patients only suffering from inflammation, 31 only suffering from nephritis and 19 suffering from both. As such, the two binary classification problems are relatively balanced: 59/120 diseased (inflammation) and 50/120 diseased (nephritis).

**Breast cancer diagnosis** (*Mangasarian, Street & Wolberg, 1995*)**.** The Wisconsin diagnostic breast cancer database (WDBC) consists of 569 cell observations, 357 of which are benign and 212 malignant, with 30 continuous features. In its original publication paper, it has been used for classification using Linear Programming with an estimated accuracy of 97.5%.

**Cardiotocography** (*Ayres-de Campos et al., 2000*)**.** This database consists of 21 variables extracted from 2,126 fetal cardiograms. Eighteen variables can be considered continuous, one categorical and two ordinal. The labelling distinguishes between 1,655 normal, 295 suspected and 176 pathological records. This study limits the problem to a binary classification (normal vs suspected/pathological).

**Chronic kidney disease** (*Rubini, Eswaran & Soundarapandian, 2015*)**.** This dataset with 400 observations and 24 variables contains many missing values, about 10% of the data. Therefore, the dataset was reduced. Firstly, the variables missing in more than 10% of the patients were eliminated. Then, the remaining patients with missing values were removed. This resulted in a dataset of 354 observations and 12 variables. Alternatively, imputation could be attempted, but that is outside the scope of the current work. Eight of the remaining variables are binary, four are continuous. The remaining observations contain 220 healthy controls and 134 patients.

**Indian liver patient data** (*Ramana, Babu & Venkateswarlu, 2012*)**.** The set contains 583 observations of nine continuous variables and one binary variable. Four observations were removed due to missing data. The goal is to distinguish 414 healthy controls from 165 liver patients.

## RESULTS AND DISCUSSION

Before presenting the full comparison of classifiers on several datasets, an example will clarify the impact that iterative reweighting can have. To this end, `lpICS` with and without reweighting was applied to the breast cancer dataset. Interactions were disabled and tuning was performed without user intervention, based on the default cutoff. As described earlier,

**Table 1  The effect of `lpICS` reweighting on the breast cancer dataset.** Complexity is reported as the number of selected variables and the total number of variable intervals. Training and test results are given as the Area Under the Curve (AUC), the accuracy (acc) and the slope and bias of the calibration curve.

| Without reweighting | | | |
|---|---|---|---|
| 23 vars | | 68 intervals | |
| Training results | | | |
| AUC | 1 | Acc | 1 |
| Slope | 1 | Bias | 0 |
| Test results | | | |
| AUC | 0.989 | Acc | 0.890 |
| Slope | 0.762 | Bias | 0.019 |
| **With reweighting** | | | |
| 2 vars | | 5 intervals | |
| Training results | | | |
| AUC | 0.958 | Acc | 0.958 |
| Slope | 1 | Bias | 0 |
| Test results | | | |
| AUC | 0.933 | Acc | 0.916 |
| Slope | 0.893 | Bias | 0.030 |

two-thirds of the data was randomly selected as training data, while the remaining third is used for assessment. Table 1 shows the results on the breast cancer dataset discussed earlier. Reweighting leads to a much simpler model. Due to its iterative nature, it allows many additional variables to be discarded. Furthermore, it also reduces the number of intervals per variable, albeit only moderately in this case: from an average of about 3 to an average of 2.5 per variable. The training and test results show that overfitting occurred without reweighting which is not surprising with the large number of variables. Training results are perfect and test results are influenced by the effect of overfitting, particularly resulting in a low calibration slope. In contrast, applying reweighting leads to a much simpler model consisting of only two variables. A comparison of the training and test results show the accuracy and calibration slope are closer together, indicating a better generalization. The test AUC is lower than without reweighting, but the accuracy has improved. The simpler model also leads to a better calibration slope, although the bias increases with 1%.

Next, the toolbox is validated on all datasets discussed earlier. Table 2 provides an overview of the performance and model complexity of all methods described in the previous sections. Performance is judged by the Area under the ROC curve (AUC). The model complexity of the ICS methods is given by the number of variables (V) and intervals (I) in the final model. Since this measure is not applicable for non-ICS methods their value has been replaced by NA. However, one can argue they use all variables in the datasets.

One can observe that the classical machine learning methods seem to outperform ICS in most cases. A statistical analysis was carried out to judge the significance of this perception. Firstly, all algorithms were compared jointly over all datasets using a non-parametric Friedman test on the AUC values, which proved significant ($p = 0.022$). Subsequently, a multiple comparison between all algorithms was carried out to identify which algorithms

**Table 2** **Comparison of test performance (AUC) and complexity (I: number of intervals, V: number of variables) for all ICS setups (`lpICS` and `enICS`, each with and without preselection), Decision Tree (DT), Naive Bayes, linear and nonlinear (RBF) SVM.** Datasets include Acute inflammation ('Inflammation' and 'Nephritis' labels), Breast Cancer Diagnosis, Cardiotocography ('Cardio'), Chronic kidney disease ('Kidney') and Indian Liver Patient data ('Liver').

| | Inflammation | | Nephritis | | Breast cancer | | Cardio | | Kidney | | Liver | |
|---|---|---|---|---|---|---|---|---|---|---|---|---|
| | AUC | I (V) | AUC | I (V) | AUC | I (V) | AUC | I (V) | AUC | I (V) | AUC | I (V) |
| lpICS | 1 | 6 (3) | 1 | 8 (4) | 0.933 | 5 (2) | 0.945 | 11 (5) | 0.932 | 4 (2) | 0.677 | 60 (9) |
| lpICS-pre | 1 | 6 (3) | 1 | 4 (2) | 0.933 | 5 (2) | 0.927 | 96 (8) | 0.959 | 8 (3) | 0.688 | 20 (5) |
| enICS | 0.962 | 4 (2) | 1 | 4 (2) | 0.942 | 4 (2) | 0.959 | 14 (6) | 0.938 | 6 (3) | 0.685 | 6 (2) |
| enICS-pre | 0.955 | 4 (2) | 1 | 4 (2) | 0.942 | 4 (2) | 0.873 | 6 (3) | 0.941 | 8 (2) | 0.685 | 6 (2) |
| DT | 1 | / (4) | 1 | / (2) | 0.936 | / (3) | 0.935 | / (13) | 0.952 | / (4) | 0.659 | / (3) |
| Naive Bayes | 1 | NA | 1 | NA | 0.976 | NA | 0.938 | NA | 0.964 | NA | 0.720 | NA |
| SVM-lin | 1 | NA | 1 | NA | 0.993 | NA | 0.957 | NA | 0.948 | NA | 0.696 | NA |
| SVM-rbf | 1 | NA | 1 | NA | 0.994 | NA | 0.957 | NA | 0.985 | NA | 0.706 | NA |

**Table 3** **Average ranks of the classifier AUC values over all datasets.** Lower values are an indication of possible better performance.

| lpICS | lpICS-pre | enICS | enICS-pre | DT | Naive Bayes | SVM-lin | SVM-rbf |
|---|---|---|---|---|---|---|---|
| 5.75 | 4.92 | 4.92 | 6.08 | 5.33 | 3.17 | 3.42 | 2.42 |

performed significantly different. The Bergmann-Hommel procedure was applied to the overall significance level of 0.05 as posthoc correction for the influence of multiple comparison (*Ulaş, Yıldız & Alpaydın, 2012*). Here, no significant differences were found, indicating we cannot reject that all algorithms perform equally well. The average ranks of all algorithms as determined in the Friedman test are displayed in Table 3. Furthermore, Fig. 7 shows a graphical representation of the multiple comparison test (*Demšar, 2006*). The dashed lines correspond to the average ranks of the corresponding algorithms on a scale of all possible ranks. Note that the ranks are inversely proportional to the AUC values: lower ranks indicate higher AUC values since the first-ranking algorithm is the best. The bold line groups all algorithms for which the null hypothesis of equal rank distribution means cannot be rejected on a pair-by-pair basis. In this case, all algorithms are in a single group since none of the pairwise tests was significant. Hence, one cannot claim that traditional Machine Learning algorithms outperform any of the ICS approaches in all cases.

In contrast, one can observe the simplicity of most ICS models which leads to its interpretability. The total number of intervals for all ICS models is usually below 10, resulting from two to four variables. One can also conclude that ICS outperforms the interpretable decision tree in terms of number of selected variables in almost all cases.

In case of the acute inflammation dataset ('Inflammation' and 'Nephritis') the labels have clearly been created using a fixed set of rules, as this set was meant to evaluate an expert system. Hence, nearly all methods manage to find a perfect model, except for the `enICS` approaches with Inflammation due to oversimplification. This case also illustrates that `lpICS` and `enICS` do not necessarily yield the same model due to the different problem formulation based on resp. classification and regression.

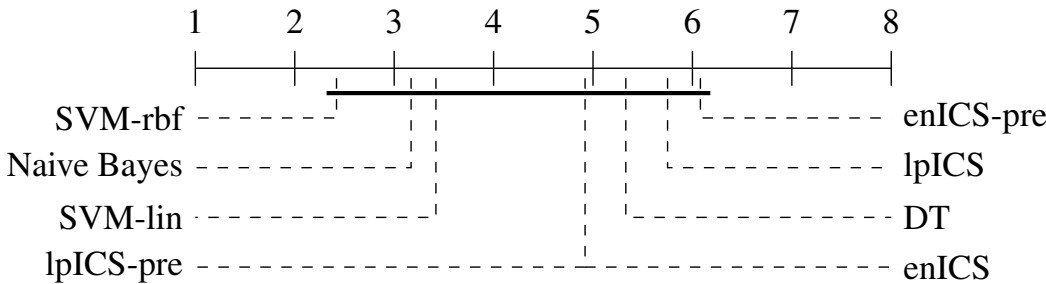

**Figure 7** **Representation of the multiple comparison of the AUC values for all algorithms using Bergmann-Hommel posthoc correction after a non-parametric Friedman test.** Dashed lines show the average rank of the corresponding algorithms over all datasets. The bold line shows where the null hypothesis of equal distribution means could not be rejected: no significant differences were found.

The results also demonstrate a shortcoming of ICS. In its automatic mode, it does not always select the trade-off one would aim for in manual training. Indeed, the automatic trade-off is performed in a rather naive way. This phenomenon presents itself at two different levels. Firstly, preselection boosts execution speed in large problems, but it does not always lead to the best model. This is particularly true for the cardiotocography problem, where `lpICS` yields a simpler model than `lpICS-pre` and yet obtains higher performance because some informative variables have nevertheless been discarded in the preselection. This is possible because preselection, although informed, still happens at the level of the complete effects (main effects or interactions). It is not as optimized as the subsequent workflow to take the interval structure into account. This drawback also exists in semi-automatic mode, but a user could base a decision on stability of the curve as well, which automatic mode does not take into account. Still, a good model is obtained, but not the simplest one. Secondly, one can observe a simpler, yet accurate model can be available. In case of the breast cancer problem, `enICS` yields a slightly simpler, yet better model. Also, in the liver dataset, one would rather be interested in the `enICS` model. However, keep in mind that the starting conditions are different for `lpICS` and `enICS`, both due to the problem formulation and due to the fact that the former considers interactions, whereas the latter does not. If the impact of the tuning parameters would be presented to the user, he or she would be able to guide the model towards the simpler, more stable and more powerful models.

The observed, yet insignificant, differences in model performance between ICS and standard ML approaches can be explained by a number of factors. Most importantly, the model is different, including its data transformation. ICS uses a piecewise linear data model but does not pose any restrictions on the weight distribution. Hence, it can model e.g., linear, quadratic or even sinusoidal effects, within the limits of its granularity. Furthermore, it has the advantage of taking the variable type into account. In a way, it transforms all variables to a categorical representation. The Decision Trees follows a similar approach and is therefore also interpretable. Its drawback is the larger number of variables selected

here. Moreover, the flat structure of ICS is easier to interpret than the nested structure of the trees: how can one easily determine or quantify the impact of a single variable?

In contrast to ICS and DT, Naive Bayes attempts to model the underlying variable distributions to make statistical inferences. In practice, its default assumption of gaussianity will probably not always be satisfied. Table 2 shows this does not necessarily result in lower performance. Moreover, it can deal with binary and categorical data via an adjustment of the distribution assumption. The linear SVM works surprisingly well given its limitations. It does not take variable types into account at all and considers only a simple linear combination of the variables. Yet this model appears to be largely sufficient for most, if not all, of the problems considered here. Finally, the nonlinear Radial Basis Function (RBF) SVM is known to be a universal approximator, that is, it can describe any model. However, whether it detects the correct one is a matter of tuning of the available data. Due to its flexible model, it is more prone to overfitting than its linear counterpart.

The validation datasets are representative for biomedical datasets in general in the fact that they do not (seem to) contain important interactions between variables. ICS only detected interactions with `lpICS-pre` for the cardiotocography set and this was far from the optimal solution, as can be derived from its comparison with e.g., `lpICS`. In all other datasets all interactions have been discarded in favour of simpler main effects. Judging from its high complexity of 60 intervals one could also suspect interactions in the Liver `lpICS` model. However, this is an example of overfitting for which nine out of ten main effects have been selected. The estimated training AUC is 0.86, which is much higher than the test AUC of 0.677 mentioned in Table 2.

Apart from the model complexity and its performance in terms of AUC, one could also be interested in its risk assessment. It can be judged by the slope and bias of the calibration curve, a (weighted) linear fit between the estimated and empirical risk. For ICS, estimated risks are available. In the Naive Bayes and DT models, posterior probabilities fulfil that role. In case of SVMs, one can build the risk assessment from the latent variables using logistic regression. A comparison of the risk prediction capacity of all algorithms is shown in Table 4. The lesser results for the Liver dataset were already apparent in Table 2, but are even more pronounced here for all classifiers excepts the SVMs.

In order to perform a statistical evaluation of the classifiers' risk prediction capabilities, the slope and bias can be combined into a single measure of goodness of fit with respect to the ideal curve. Consider the calibration curve $t = ap + b$ with slope $a$ and bias $b$ which models the relation between predicted risks $p$ and empirical ('true') risks $t$. The ideal model would be $t = 1p + 0$ with slope 1 and bias 0. Hence, one can define a calibration error measure $e$ as the $L_p$ norm of the difference between the actual and the ideal curve (*Callegaro, Pennecchi & Spazzini, 2009*). For the simplest case ($p = 1$), this yields:

$$e = \int_0^1 |(a-1)p + b| \, dp.$$

Using the error values calculated for all (slope, bias) pairs in Table 4 an identical statistical analysis as before can be carried out. Table 5 and Fig. 8 show the results. The Friedman test was significant ($p = 0.021$) and the multiple comparison test indicated a single significant

**Table 4** Comparison of the quality of risk estimation on the test data via the slope and bias of the calibration curve for all ICS setups (`lpICS` and `enICS`, each with preselection), Decision Tree (DT), Naive Bayes, linear and nonlinear (RBF) SVM. Datasets include Acute inflammation ('Inflammation' and 'Nephritis' labels), Breast Cancer Diagnosis, Cardiotocography ('Cardio'), Chronic kidney disease ('Kidney') and Indian Liver Patient data ('Liver').

| | Inflammation | | Nephritis | | Breast Cancer | | Cardio | | Kidney | | Liver | |
|---|---|---|---|---|---|---|---|---|---|---|---|---|
| | Slope | Bias | Slope | Bias | Slope | Bias | Slope | Bias | Slope | Bias | Slope | Bias |
| lpICS | 1 | 0 | 1 | 0 | 0.89 | 0.03 | 0.95 | −0.01 | 0.95 | 0.05 | 0.44 | 0.40 |
| lpICS-pre | 1 | 0 | 1.02 | −0.02 | 0.89 | 0.03 | 0.92 | 0.01 | 0.98 | 0.02 | 0.55 | 0.32 |
| enICS | 1.01 | −0.01 | 1.02 | −0.02 | 0.96 | −0.01 | 1.04 | −0.01 | 0.95 | 0.05 | 0.72 | 0.21 |
| enICS-pre | 1 | 0.01 | 1.02 | −0.02 | 0.96 | −0.01 | 1.04 | 0 | 0.92 | 0.05 | 0.72 | 0.21 |
| DT | 1 | 0 | 1 | 0 | 0.90 | 0.01 | 0.94 | 0.03 | 0.91 | 0.09 | 0.63 | 0.27 |
| Naive Bayes | 1.15 | −0.05 | 1.03 | 0.05 | 0.84 | 0.05 | 0.74 | 0.06 | 0.84 | 0.18 | 0.34 | 0.61 |
| SVM-lin | 1 | 0 | 1 | 0 | 0.96 | 0.01 | 1 | 0 | 0.99 | 0.03 | 0.72 | 0.22 |
| SVM-rbf | 1 | 0 | 1 | 0 | 0.94 | 0 | 0.88 | 0.03 | 0.97 | 0.03 | 0.68 | 0.24 |

**Table 5** Average ranks of the classifier calibration errors over all datasets. Lower values are an indication of possible better performance.

| lpICS | lpICS-pre | enICS | enICS-pre | DT | Naive Bayes | SVM-lin | SVM-rbf |
|---|---|---|---|---|---|---|---|
| 4.83 | 4.83 | 4.00 | 4.00 | 4.17 | 8.00 | 2.83 | 3.83 |

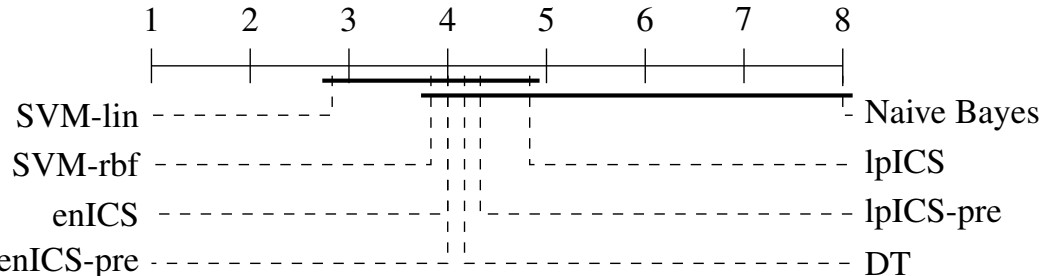

**Figure 8** Representation of the multiple comparison of the calibration error for all algorithms using Bergmann-Hommel posthoc correction after a non-parametric Friedman test. Dashed lines show the average rank of the corresponding algorithms over all datasets. The bold line shows where the null hypothesis of equal distribution means could not be rejected. Two groups are identified: one excluding SVM-lin and one excluding Naive Bayes since these two had significantly different ranks.

difference. We can reject the hypothesis that Naive Bayes' calibration is equally good as SVM-lin's. As a result, Fig. 8 has two groups for which equal calibration performance cannot be rejected.

Despite the lack of significant differences, one can make some observations on part of the datasets. Although Naive Bayes explicitly works with probability distributions it tends to be similar or weaker than the ICS models. This is in particular clear in the cardiotocography and kidney disease set. This might be caused by a large deviation of the actual data distribution compared to the assumed gaussianity. More generally, it is ranked last for every dataset.

The SVMs provide a good prediction of the risks, particularly if the linear model is used as can be judged from its first rank. For the difficult Indian liver dataset, SVM-lin still offers a good model. However, it is matched by `enICS`, but due to the model restrictions it has a slightly lower AUC.

Once again, one can conclude that, in general, performance loss is not significant, especially considering the gain in interpretability. ICS sometimes even proves to be the better choice, as can be judged from its slightly superior AUC and very similar risk estimation quality for the cardiotocography set. Furthermore, it is evident that one could use ICS for simple problems without any penalty, as seen in the acute inflammation datasets (inflammation and nephritis). Even better results can be obtained when tuned in a semi-automatic way.

## CONCLUSION

This paper presented the Interval Coded Scoring Toolbox for MATLAB. It extracts scoring systems mainly aimed at clinical and medical environments, but with benefits for all fields in which interpretability is a key requirement. These systems identify important effects and interactions in a dataset and explain how they contribute to the risk for a specific phenomenon. In contrast to many common medical scoring systems it is firmly rooted in optimization theory, offering both a Linear Programming-based approach and one depending on elastic nets and SVMs. The best results are obtained by making a good trade-off between model complexity and performance. For this reason, the impact of tuning is presented to the user who can intervene at several points in the training process. The main goal of the ICS model is its interpretability, with limited concessions to the model performance. Therefore, the toolbox not only helps to state a new ICS problem and train the model, but it also provides a colour-coded visualization, including various evaluation metrics.

The toolbox was described, highlighting that a new model can be obtained by writing just a few lines of MATLAB code. The validation study showed that ICS often seems to be outperformed by common techniques such as SVMs in terms of performance and risk estimation but these differences could not be proven significant. In particular, no difference could be observed for very simple problems. Furthermore, for more complex problems, the difference is balanced with the fact that the ICS model is easily interpretable and applicable to new data, even without the need for a computer.

The toolbox has been made available free of charge for academic use with the conviction that it will be a valuable tool for all fields requiring interpretable decision support.

## ACKNOWLEDGEMENTS

This paper reflects only the authors' views and the European Union is not liable for any use that may be made of the contained information.

### Funding

This work was supported by the Bijzonder Onderzoeksfonds KU Leuven (BOF): it is part of the SPARKLE project (Sensor-based Platform for the Accurate and Remote monitoring of Kinematics Linked to E-health #: IDO-13-0358). It is also supported by imec funds 2017 and imec ICON project. The research leading to these results has received funding from the European Research Council under the European Union's Seventh Framework Programme (FP7/2007-2013)/ERC Advanced Grant: BIOTENSORS (no. 339804). The funders had no role in study design, data collection and analysis, decision to publish, or preparation of the manuscript.

### Grant Disclosures

The following grant information was disclosed by the authors:
Bijzonder Onderzoeksfonds KU Leuven—SPARKLE project: IDO-13-0358.
imec funds 2017.
imec ICON projects.
European Research Council: (FP7/2007-2013).
ERC Advanced Grant—BIOTENSORS: 339804.

### Competing Interests

Lieven Billiet and Sabine Van Huffel are affiliated with imec Leuven through a research collaboration.

### Author Contributions

- Lieven Billiet conceived and designed the experiments, performed the experiments, analyzed the data, prepared figures and/or tables, performed the computation work, authored or reviewed drafts of the paper, approved the final draft.
- Sabine Van Huffel conceived and designed the experiments, authored or reviewed drafts of the paper, approved the final draft.
- Vanya Van Belle conceived and designed the experiments, performed the computation work, authored or reviewed drafts of the paper, approved the final draft.

### Data Availability

The code toolbox is restricted by the KU Leuven university to academic use, as stated on the webpage of the toolbox. All non-academic use of the toolbox is only possible after explicit institutional permission. Interested academic users can get access at the KU Leuven institutional database: https://esat.kuleuven.be/stadius/icstoolbox.

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
