# Peer review of "Interval Coded Scoring: a toolbox for interpretable scoring systems"

_PeerJ Computer Science, doi:10.7717/peerj-cs.150_

## Round 0.1 · original submission · Major Revisions

Please prepare a new manuscript attending all the suggestions provided by the reviewers.

In order to consider the statistical treatment suggested by reviewer 2, you could find useful the information available at the following address:

http://sci2s.ugr.es/sicidm

Reviewer 1 ·

Basic reporting

No comment

Experimental design

No comment

Validity of the findings

No comment

Additional comments

In this manuscript the authors introduce a Matlab toolbox for ICS. This toolbox provides not only the basic code to perform ICS but also an interactive process with plots and guides to help the users.

This work is well structured, complete and self contained. The toolbox validation section is a plus encouraging the reader to use this new toolbox.

Some minor comments to improve reading of the manuscript:

* In page 4 the authors indicate the possibility of coding interactions. I would recommend adding a sentence offering an example of an interaction in a medical domain.
* In the next page the authors talk about the reweighting process. An example of a solution of ICS with and without this process could help the reader understand why interpretation is a must in some domains.
* A link to the software or directions on where to obtain, public repository, etc. of the toolbox is mandatory.

Reviewer 2 ·

Basic reporting

There are several use of parenthesis which can be misunderstood. For example, at line 23 of abstract, the authors write “...for a (binary) classification problem...” which can be understood as the model can be applied in binary and non-binary classification problems. However, along the paper the authors use the binary classification. I suggest to clarify it with examples or eliminating the parenthesis.

Experimental design

Some suggestions are made in the attachment.

Validity of the findings

- In section 3, there are some comparisons with classical machine learning methods. Concretely, at line 465 the authors say “… it is clear that the differences are relatively small...”. It is a subjective fact without any other supports. This sentence should be supported with some objective studies or analysis, such us,non-parametrical analysis (e.g. Shaffer test.)
- In order to achieve a better comparison, the authors should add the ranking data and carry out an analysis of them, among their proposals and among the rest of them.

Annotated reviews are not available for download in order to protect the identity of reviewers who chose to remain anonymous.

---

## Round 0.2 · accepted · Accept

After checking that authors have addressed all the reviewers' suggestions, I think the paper is ready for publications. Congratulations,